# In situ analysis of the bulk and surface chemical compositions of organic aerosol particles

Yuqin Qian[1], Jesse B. Brown[1], Zhi-Chao Huang-Fu[1], Tong Zhang[1], Hui Wang[1,2], ShanYi Wang[1,3], Jerry I. Dadap[4] & Yi Rao ⬛[1✉]

Understanding the chemical and physical properties of particles is an important scientific, engineering, and medical issue that is crucial to air quality, human health, and environmental chemistry. Of special interest are aerosol particles floating in the air for both indoor virus transmission and outdoor atmospheric chemistry. The growth of bio- and organic-aerosol particles in the air is intimately correlated with chemical structures and their reactions in the gas phase at aerosol particle surfaces and in-particle phases. However, direct measurements of chemical structures at aerosol particle surfaces in the air are lacking. Here we demonstrate in situ surface-specific vibrational sum frequency scattering (VSFS) to directly identify chemical structures of molecules at aerosol particle surfaces. Furthermore, our setup allows us to simultaneously probe hyper-Raman scattering (HRS) spectra in the particle phase. We examined polarized VSFS spectra of propionic acid at aerosol particle surfaces and in particle bulk. More importantly, the surface adsorption free energy of propionic acid onto aerosol particles was found to be less negative than that at the air/water interface. These results challenge the long-standing hypothesis that molecular behaviors at the air/water interface are the same as those at aerosol particle surfaces. Our approach opens a new avenue in revealing surface compositions and chemical aging in the formation of secondary organic aerosols in the atmosphere as well as chemical analysis of indoor and outdoor viral aerosol particles.

[1] Department of Chemistry and Biochemistry, Utah State University, Logan, UT 84322, USA. [2] Department of Chemistry, Fudan University, Shanghai 200433, China. [3] Department of Physics and Astronomy, Barnard College, New York, NY 10027, USA. [4] Stewart Blusson Quantum Matter Institute, University of British Columbia, Vancouver, BC V6T 1Z4, Canada. ✉email: yi.rao@usu.edu

Understanding the chemical and physical properties of aerosol particles is an important scientific, engineering, and medical topic that is crucial to air quality, human health, and environmental chemistry, in particular indoor transmission related to viral aerosol particles[1–10]. Bio- and organic-constituted aerosols are of multiple phases, consisting of liquid, solid, or mixed particles[1]. The growth of these different types of particles is intimately related to physical properties and chemical reactions in the gas and particle phases, and at the gas/aerosol particle interface[1–8]. Atmospheric particles interact with gas-phase chemical species in a range of heterogeneous and multiphase processes, from the chemical processing of surface organics by gas-phase oxidants to the hygroscopic uptake of water and even cloud droplet activation[3,11]. Particle surfaces serve as the gateway for all of these interactions. The structure, mass transport kinetics, molecular-level dynamics, and heterogeneous chemical reactions are different from those of their corresponding bulk in the surface regions of an aerosol particle[9,11,12]. For example, the ability of environmentally relevant surfaces that have acidic properties to mediate surface chemistry has important implications for the fate of many environmental species. Surface pH of aerosol particles is believed to alter the heterogeneous chemistry, optical properties, and ice and cloud nucleating ability of atmospheric aerosols[13,14]. Despite the importance of aerosol surfaces, direct evidence is needed to reveal chemical structures, binding or adsorbate behaviors, kinetics, dynamics, and heterogenous reactions of aerosols.

The physical and chemical properties of the aerosol particle surfaces are different from those of their in-particle phase[3,8,12,15–26]. Recent studies showed that biological and organic materials prefer to remain at aerosol surfaces and occupy surface sites of aerosol particles[3,8,15,17,27–31]. The biological and organic materials at the particle surface can change chemical composition of the particle by manipulating in and out of its surface, surface chemistry, physical properties, ands on[8,15,32]. More importantly, the surfaces of aerosol particles are expected to host unique physical and chemical settings for biological and chemical reactions. In the atmosphere, chemical structures and reactions of constituents at aerosol surfaces are driven by the photochemical production of reactive gas-phase species, such as ozone and hydroxyl radicals[9,33–38]. As a result, these oxidizing agents react at the surface of particles with organic species in processes which are heterogeneous. On the other hand, the surface properties of these particles play a significant role in the kinetics of reactive uptake and the subsequent chemical reactions[9,11,12]. These heterogeneous reactions further cause changes in the composition and physical properties of aerosols. The questions arise from how particle surfaces play a role in these reactions for liquid or solid particles. Theoretical work has predicted that interfaces alter bulk chemical equilibria and accelerate reactions in micro-compartments[39]. There has been a lack of analytical techniques capable of investigating the surface chemistry of submicron aerosols directly. As a result, surface-specific aerosol chemical processes and their characteristics have only been inferred based on indirect observations and modeling.

Previous studies of aerosol surfaces are mainly focused on ex situ measurements of aerosol particles collected from the field[40–43]. Our recent attempts have demonstrated in situ measurements of molecular behaviors at aerosol particle surfaces by developing second harmonic scattering (SHS) and electronic sum frequency scattering (ESFS) techniques[44–47]. These measurements provide direct evidence of organic molecules existing at aerosol surfaces, surface populations of the organic molecules, surface polarity, as well as configurations of the organic molecules at aerosol surfaces. Although these preliminary results offer direct measurements of physical properties at aerosol surfaces, SHS and

ESFS techniques cannot provide chemical information of organic species.

Our primary purpose is to identify organic species and their surface properties of aerosol particles floating in the air directly. In this work, we present direct probing of chemical species at surfaces of laboratory-generated aerosol particles in real time by developing vibrational sum frequency scattering (VSFS) spectroscopy. Furthermore, we compare the concentration-dependent adsorption of organic species to both planar and spherical particle surfaces.

## Results and discussion

**Simultaneous detection of vibrational structures of propionic acid at particle surfaces and in particle phase.** Experimental details and precise descriptions of the depictions in Fig. 1 can be found below. To identify chemical structures of molecules at aerosol particle surfaces, VSFS experiments were implemented. Figure 2a presents an HHH-polarized VSFS spectrum of aerosol particles produced from 4.0 M propionic acid in 0.5 M NaCl solution when both the picosecond 1025 nm and the 3425 nm IR beam were overlapped spatially and temporally. Three main peaks appear at 2887.3, 2950.9, and 2991.8 $cm^{-1}$ for the propionic acid aerosols, which were attributed to the symmetric stretching mode of $-CH_3$ ($CH_3$-ss), the Fermi resonance between the symmetric $-CH_3$ stretching mode and overtones of $-CH_3$ deformation vibrations (Fermi), and the asymmetric stretching mode of $-CH_3$ ($CH_3$-as), respectively[48].

To reveal vibrational structures of molecules in the particle phase, HRS spectra were taken for aerosol particles from the same stock solution by setting a grating centered at 605 nm. Figure 2a shows an HHH-polarized HRS spectrum with three main peaks at 2873.0, 2933.6, and 2986.1 $cm^{-1}$ from aerosol particles from the same seed solution of 4.0 M propionic acid mixed with 0.5 M NaCl. Based upon energy conservation, the peaks originated from $2\omega_{1027nm}$ to $\omega_{3425nm}$ and were assigned to be hyper-Raman modes for $CH_3$-ss, Fermi, and $CH_3$-as of propionic acids in solution. It is interesting to notice that the relative intensities of the three peaks in the HRS spectrum are different from those for VSFS. The VSFS signal for $CH_3$-ss is larger than that for $CH_3$-as, while the HRS signal for $CH_3$-ss is weaker than the corresponding $CH_3$-as. Such differences might come from random orientations of propionic acid molecules in the particle phase.

To further examine the origin of surface VSFS and bulk HRS of aerosol particles, concentration-dependent experiments were carried out. Figure 2b compares VSFS and HRS signals for the $CH_3$-as mode as a function of bulk concentration of propionic acid. As expected, HRS intensity is linearly proportional of the bulk density[49]. On the other hand, VSFS shows a different behavior with the concentration of propionic acid in a nonlinear manner, as observed in ESFS and SHS before[44–47]. These results further confirmed that VSFS signal comes from the surface, instead of the bulk. Additionally, experiments were conducted to compare the intensities of VSFS signals as they relate to the particle number density. By diluting the aerosol stream with humidified $N_2$ gas to mitigate evaporation, we found that VSFS signal intensity is linearly proportional to particle density, as shown in Fig. 2c, alligning with that for SHS[44].

**Polarized VSFS spectra of fatty acids at aerosol surfaces.** To demonstrate orientational configurations of molecules, we measured different polarized VSFS spectra at aerosol surfaces. Figure 3a displays four polarized VSFS spectra of aerosol surfaces from 4 M propionic acid, including HHH, VVH, VHV, and HVV. Only HHH and VVH polarizations show VSFS signals, while no signals were observed for the VHV and HVV. The Fermi

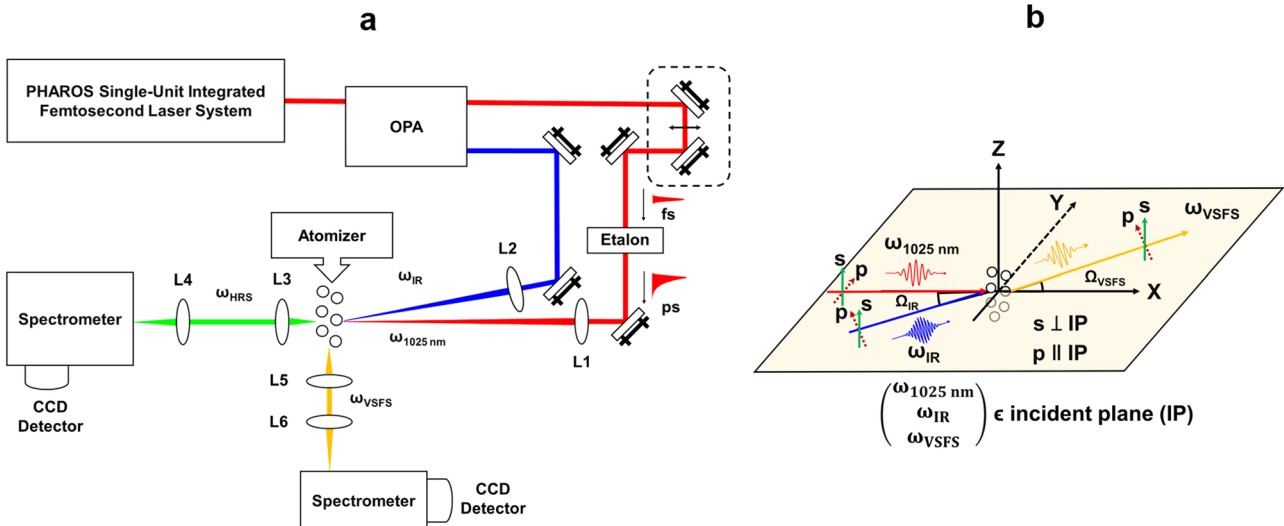

**Fig. 1 Schematic diagram for vibrational sum frequency (VSFS) and hyper-Raman scattering (HRS) experiments. a** Schematic diagram of both the VSFS and HRS experiments. A spectrally narrow picosecond 1025 nm from an etalon was combined with a broadband OPA-generated IR femtosecond pulse at the aerosol samples to yield sum-frequency and hyper-Raman signals. L1: 1″ achromatic lens of a 25 cm focal length. L2: 1″ CaF$_2$ lens of a 15 cm focal length; L3/L5: 2″ achromatic lens of a 3.2 cm focal length, and L4/L6: 2″ achromatic lens of a 7.5 cm focal length. **b** The narrow picosecond 1025 nm and the broadband IR light were non-collinearly incident on samples at angle of $\Omega_{1025} = 0°$ and $\Omega_{IR} = 5°$ with respect to the X axis. The incident plane and scattering plane were in the X-Y plane, with the Z axis perpendicular to the optical table.

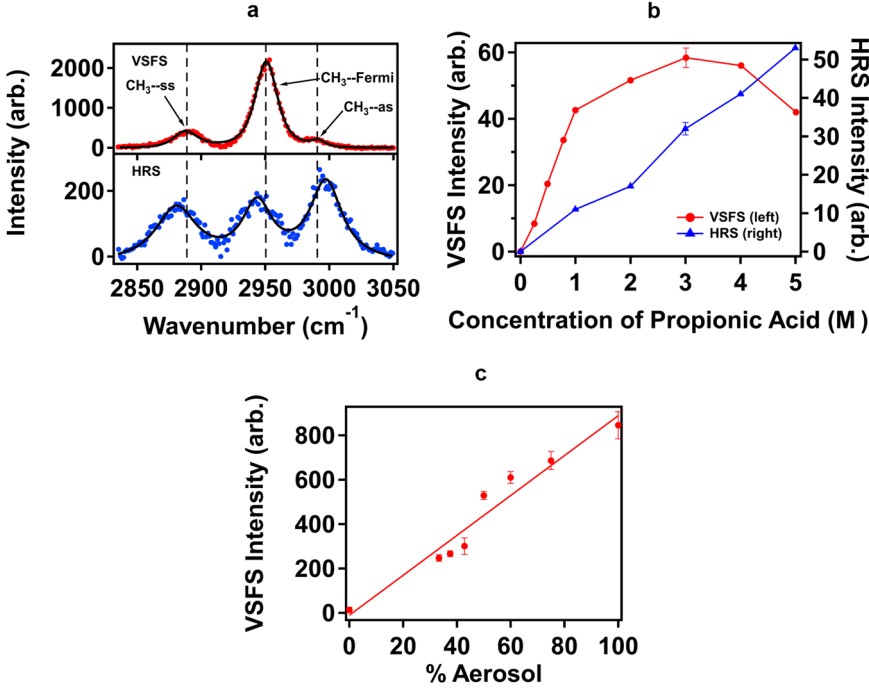

**Fig. 2 Spectra and isotherms of vibrational sum frequency (VSFS) and hyper-Raman scattering (HRS) experiments. a** HHH-polarized (H = horizontal polarization) configuration spectra for VSFS (upper) and HRS (lower) for aerosol particles from a 0.5 M NaCl seed solution with 4.0 M propionic acid. **b** VSFS (red solid circle, left axis) and HRS (blue solid triangle, right axis) intensities at 2991.8 cm$^{-1}$ (CH$_3$-as) as a function of propionic acid concentrations added. Error bars represent 5% deviation. **c** Relationship of aerosol particle density to VSFS intensity. Error bars represent 1 standard deviation.

peak of –CH$_3$ shows the largest signal, as compared with those for CH$_3$-ss and CH$_3$-as for all polarizations. The CH$_3$-as almost disappears for VVH. These results suggest that molecules are oriented with certain ordering at the aerosol particle surfaces in Fig. 3b, resulting in silent peaks in the two polarization combinations.

To reveal quantitative molecular orientation of propionic acids at aerosol surfaces, we considered two non-vanishing hyperpolarizabilities of the –CH$_3$ group: $\beta_{ccc}^{(2)}$ and $\beta_{aac}^{(2)} = \beta_{bbc}^{(2)} = R\beta_{ccc}^{(2)}$ for the symmetric stretching mode, and $\beta_{aca}^{(2)}$ for the asymmetric stretching mode. The $R$ is the ratio of $\beta_{aac}^{(2)} = \beta_{bbc}^{(2)}$ and $\beta_{ccc}^{(2)}$ with a value of 3.0 for –CH$_3$ for the acids in our case[50]. Previous studies showed that the Mie scattering theory may be applied to obtain orientations of chemical groups at particle surfaces in vibrational sum frequency scattering by Roke and co-

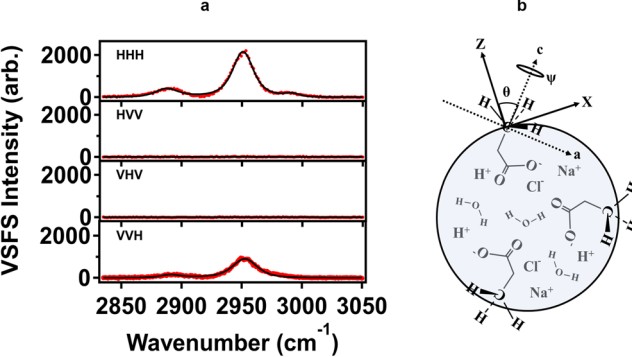

**Fig. 3 Vibrational sum frequency scattering (VSFS) spectra of aerosol particles with different polarizations. a** Polarized VSFS spectra for aerosol particles from 4 M propionic acid in a 0.5 M NaCl seed solution with HHH, HHV, HVH, and VVH (H horizontal polarization, V vertical polarization). **b** Schematic of orientational configuration of the –CH₃ group of propanoic acids at the aerosol surface.

workers[51]. We have attempted to apply this excellent theory to propionic acid for orientational behaviors of –CH₃ in vibrational sum frequency scattering. We have correlated the hyperpolarizabilities of the chemical group with the second-order susceptibilities in the framework of the surface coordinate system[50,52]. We have also considered distribution function of the orientation of the chemical group[53,54]. One result from the theory was that our polarized spectra were qualitatively consistent with the selection rules therein. In other words, VSFS for VHV and HVV are not allowed while those for HHH and VVH are allowed. Regrettably, this theory fails to quantitatively explain either of the polarized VSFS signals for the symmetric and asymmetric modes of –CH₃ groups of propionic acid molecules at aerosol particle surfaces. Our experimental results showed that the VSFS signal for the HHH polarization is always larger than that for the VVH polarization, while the theory predicted the opposite. This may be due to the fact that refractive index mismatching contributions cannot be neglected and the scattering of small particle might be considered in our case[55]. These quantitative analyses will be our next endeavor.

**Adsorption behaviors of molecules at aerosol particle surfaces.** To compare surface adsorption behaviors of propionic acid at the aerosol particle surface with those at the air/water interface, concentration dependent VSFS and VSFG experiments, namely isotherms, were carried out. Figure 4a compares VSFS and VSFG intensities for the CH₃-as mode as a function of bulk concentration of propionic acid in 0.5 M NaCl. Both isotherms show a nonlinear relation with the concentration of propionic acid; further confirming that molecular behaviors from surface-specific VSFS are different from those from bulk-dominated HRS.

To quantitatively examine surface adsorption ability of propionic acid, we considered the Langmuir model for molecular behaviors at the two surfaces. The VSFS or VSFG electric field is proportional to surface density, $N_S$. The surface electric field, $E_S$, is expressed as[47,56]

$$E_S \propto N_S \propto \frac{Kc}{Kc + 55.5} \qquad (1)$$

where $K$ is the surface adsorption constant at equilibrium and $c$ is the concentration of propionic acid in the seed solution.

We first compared the surface adsorption free energy of propionic acid at the aerosol surface with that at the air/water interface. Fittings of the model in Eq. 1 to the two curves in Fig. 4a generate the surface adsorption constants for propionic acid of

$1.67 \pm 0.48 \times 10^2$ at the aerosol surface, and $4.95 \pm 0.53 \times 10^2$ at the air/water interface, yielding adsorption free energy values of $-12.69 \pm 0.28\ kJ\ mol^{-1}$ and $-15.38 \pm 0.11\ kJ\ mol^{-1}$, respectively. It is surprising that the adsorption ability of the molecules at the curved particle surface is lower than that at the planar surface. Although the large difference in adsorption constants is effectively lost in the logarithmic free energy function for propionic acid, this difference may be more dramatic when considering other adsorbates.

We further compared surface adsorption free energies of different fatty acids at the aerosol surface by carrying out concentration dependent VSFS experiments for butanoic acid as well. Figure 4b displays VSFS signals for different concentrations of butanoic acid for the CH₃-as mode. The surface adsorption free energy was estimated to be $-15.96 \pm 0.24\ kJ\ mol^{-1}$ for butanoic acid at the aerosol surface. As expected, the longer the carbon chain, the more negative the surface adsorption free energy.

## Conclusions
We have presented in situ analysis of chemical structures of organic species at aerosol surfaces by developing and implementing vibrational sum frequency scattering spectroscopy. This unique setup also offered simultaneous chemical structure analysis of species by hyper-Raman scattering from the in-particle phase. Such a surface-specific structural probe allowed us to implement structural analysis and observe orientational configuration and surface adsorption behaviors of molecules at aerosol surfaces in real time. Our qualitative results showed that molecules are oriented with certain orientations at aerosol surfaces. These results of the spatial configurations are important for determining rates and yields of subsequent chemical reactions occurring at aerosol particle surfaces. Surface adsorption free energy of propionic acid at aerosol surfaces is less negative than that at the air/water interface. These results imply that previous modeling of heterogeneous atmospheric chemistry by using data from the planar surface must be re-evaluated.

The in situ VSFS method offers a direct approach for understanding the role of aerosol particle surfaces in real time. The development of the in situ simultaneous VSFS and HRS technique could also enlighten mechanistic studies of chemical reactions at aerosol surfaces and in-particle phases at the same time. Thus, we will be able to differentiate surface from bulk contributions to the production of large organic compounds in aerosol particles. These advancements will help us further understand the growth of secondary organic aerosols with the structural information of both aerosol surfaces and in the particle phase, which is beneficial to atmospheric chemistry modeling at both a regional and a global scale. Furthermore, chemical identifications at the aerosol surfaces also have significant implications in the detection of indoor and outdoor transmission of aerosol particles.

## Experimental section
**In situ vibrational sum frequency scattering (VSFS) and hyper-Raman Scattering (HRS) measurements.** Figure 1a shows a schematic setup for both the VSFS and HRS experiments. The experimental setup consisted of three main units: light sources, signal detection, and laboratory-generated aerosol particles.

*Light sources.* A femtosecond amplifier laser system (PHAROS, Light Conversion) running at a repetition rate of 100 kHz and was an average output of 8 W was used to pump an optical parametric amplifier (ORPHEUS-ONE, Light Conversion), yielding a tunable middle IR light beam from 2500 nm to 4500 nm. The residual component of the pump was exploited to

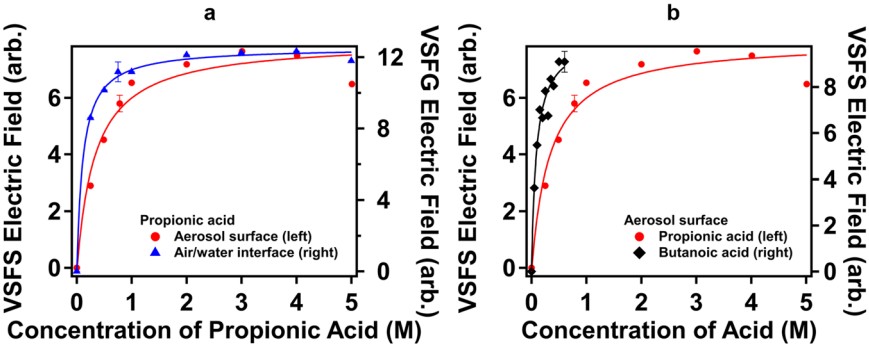

**Fig. 4 Concentration dependent adsorption to curved and planar surfaces. a** Vibrational sum frequency scattering (VSFS) electric fields of $CH_3$-as for aerosol curved surfaces (red solid circle, right axis) and VSFG electric fields of $CH_3$-as for the air/water planar interface (blue solid triangle, right axis) as a function of propionic acid concentration in a 0.5 M NaCl seed solution. **b** VSFS electric fields of –$CH_3$-as as a function of butanoic acid concentration in 0.5 M NaCl seed solution (black sold diamond, right axis), in comparison with those for propionic acid (red solid circle, left axis). Error bars represent 5% deviation.

generate a spectrally narrow picosecond pulse with a spectral linewidth of $8\,cm^{-1}$. This was achieved by inserting an etalon (SLS Optics) to filter the broadband femtosecond 1025 nm laser down to the spectrally narrow picosecond beam.

The picosecond (1025 nm) and broadband IR light beams were non-collinearly incident on samples at angles of 0° and 5°, respectively, relative to the X axis in Fig. 1b, which is similar to that for in situ ESFS[47]. Vibrational SFS, together with the two incident beams, constitutes an incident plane parallel to the optical table, defined as the XY plane. Pulse energies of 6.0 µJ and 2.0 µJ were applied for the 1025 nm and IR beams, respectively. Two lenses of 25 cm and 10 cm focal lengths were used for the 1025 nm laser and IR light beam to focus on a flow of aerosol particles. It turned out that the diameters of the focal spot were ~90 µm for the 1025 nm laser and ~80 µm for the IR beam. The polarizations of the 1025 nm and the IR beam were varied by two independent half-wave plates, and thin-film polarizers were used to select the polarization of the scattering signals. Here we denote the polarizations which are perpendicular to and parallel with the optical table as V- and H-polarized, respectively.

*Simultaneous Detection of VSFS and HRS signals.* A lens (2", $f = 3.2\,cm$) was placed at an angle of 90° relative to the X axis so that VSFS signals were collected with a wide angle (2θ) of ~60 degrees. One more lens (2", $f = 7.5\,cm$) was used to focus the collected signal on the entrance of a spectrometer in an effort to achieve spectral resolution of VSFS. A spectrometer (Acton 300i, Princeton Instruments) was combined with a charge-coupled device (Princeton Instruments, LN/CCD-1340/400) for VSFS detection. The integration time was 180 s for each VSFS spectrum at $\omega_{VSFS} = \omega_{1025nm} + \omega_{IR}$. The same collection lens as those in VSFS were placed at an angle of 0° for HRS signal, as shown in Fig. 1a. Another spectrometer (Acton 300i, Princeton Instruments) equipped with a charge-coupled device (Princeton Instruments, LN/CCD-1340/400) was used for HRS detection. The integration time for each HRS spectrum at $\omega_{HRS} = 2\omega_{1025\,nm} - \omega_{IR}$ was 60 s. As such, we were able to simultaneously obtain surface and bulk information from VSFS and HRS measurements of aerosol particles.

*Laboratory-generated aerosol particles.* A stock solution of 0.5 M NaCl was prepared as a seed for particle generations in our experiments[44–47]. Aerosol particles were produced by a constant output atomizer (TSI 3076) under a constant pressure of 40 psi and flow of 4 slpm (MKS). The density of particles was estimated

to be ca. $3.8 \times 10^6\,cm^{-3}$, with a diameter centered at near 40 nm, and size distribution spanning from 10 nm to 300 nm (TSI, Optical Particle Sizer 3330 and Nanoparticle Sizer 3910). The aerosol particles were introduced to an enclosed chamber for experiments, and an exhaust pump was used to guarantee no accumulation of aerosol particles in the chamber.

As with previous SHS experiments[44], the relationship of VSFS intensity and aerosol particle density was investigated. The $3.8 \times 10^6\,cm^{-3}$ particle density was diluted with an interjecting variable flow of $N_2$ which passed through a distilled water bubbler which intercepted the PTFE tube carrying the aerosol particles. All aerosol and gas handling were done through 1/4" OD x 11/64" ID PTFE tubing and a glass elution tip of the same diameters at the sample location. Using an additional flow controller, (MKS) the aerosols containing 4 M propionic acid in 0.5 M NaCl were diluted ranging 0–100% of the original flow.

*Planar Vibrational SFG (VSFG).* A different femtosecond laser amplifier system (UpTek Solutions) with a fundamental wavelength of 800 nm and repetition rate of 1 kHz was used for VSFG experiments at the air/liquid interface. This 800 nm pulse was used to pump an OPA (TOPAS, Light Conversions) which generated a 3250 nm IR pulse of 5 µJ. The residual 14 µJ, 800 nm pulse was directed through a translation stage to control the temporal overlap of the two pulses, followed by an air-spaced etalon (SLS Optics) and half-wave plate. A dichroic mirror was used to direct the IR and 800 nm pulses colinearly to the sample. A reflection geometry was employed in which the VSFG signal was collected by a lens and passed through a thin film polarizer before being focused on the slit of a spectrometer (Acton, SpectraPro 2300i, Princeton Instruments) fitted with a CCD detector (Princeton Instruments, LN/CCD-1340/400)[57]. Samples were contained in a 2" diameter PTFE dish placed on a rotation stage and prepared identical to those for VSFS experiments.

*Chemicals.* Propionic and butanoic acids (Acros Organics) were used as received. Ultrapure water of 18.2 MΩ·cm was used to prepare stock solutions in our experiments. NaCl salt (Fisher Chemical) was baked at 600 °C for 10 h prior to use. The acids were added to 0.5 M NaCl stock solution for desired concentrations in both VSFS and HRS experiments.

## Data availability
The datasets generated and/or analyzed during the current study are available from the corresponding authors on reasonable request.

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

## Acknowledgements

This material is based upon work supported by the National Science Foundation under Grant No. [2045084].

## Author contributions

Y.R. and Y.Q. conceived and designed experiments. Y.R. managed this project. Y.Q., J.B.B., Z-C.H-F., and T.Z. constructed and conducted experiments J.I.D., H.W., and S-Y.W. conducted theoretical considerations. Y.R., Y.Q., J.B.B., and Z-C.H-F. wrote the manuscript.

## Competing interests

The authors declare no competing interests.
