## [Peer Review File · Communications Chemistry]

Reviewers' comments:

Reviewer #1 (Remarks to the Author):

The authors report what could be an interesting technical result if they had experimental proof that their signal is indeed due to SFS from aerosol particles. Yet, given the small differences in the isotherms for the two systems studied, which is expected from consideration of the Kelvin effect, this reviewer is not convinced that even if the data presented were proven to be indeed due to SFS from aerosol particles, it would provide new physical insights.

The authors have omitted even the most basic controls to show that their signal could be from aerosol particles. If this chamber is the one that was used in the authors' prior SHG work, then the long focal lengths of their setup could easily lead to largely dominating (or even sole) contributions from the optical elements in the chamber. This lack of basic controls indicates the work is highly premature.

Below are points needed for a convincing story:

There is almost no experimental detail given, except for the laser system, detector, and lenses, so that needs to be provided, with detailed drawings. What is the sample chamber composed of? What are the exact dimensions? Where are the windows? How thick and large are they? How is the exhaust connected? How is it guaranteed and validated that no material builds up in the chamber and on the windows?

There are more fundamental questions regarding the reported results:

Were there any concentration dependent lineshape changes in the VSFS and HRS signal?

Were HRS bulk signals comparably low for butanoic acid solutions - neither one of these molecules should be very surface active, given their K_{sp} 's?

Is there any reason for the nonmonotonic trend in VSFS intensity for the propionic acid samples around 3M concentration?

Were any other spectra taken at different polarizations for the butanoic acid - this section generally could use some more explanation about the discrepancies encountered.

What are the particle concentrations and sizes for the different systems, and how were those quantified?

What are the uncertainty estimates on the adsorption isotherms, esp. in Fig 4A?

The key control needed is the one that shows the SFS signals appearing and completely disappearing when the aerosol flow is turned on and off. Even then, additional controls are needed to show zero material uptake on the optical elements in the focused portion of the time- and space-overlapped beam paths.

The second crucial control is the number density dependence of the SFS signal intensity. The authors know how important these are from their pioneering work on particles at Columbia - it's not clear why these important controls and tests were omitted here.

Even if all of these questions are eventually answered, and even if proof of true aerosol SFS is provided, the ultimate problem with the work is that the new physical insight is unfortunately lackluster in that the adsorption of the two probe species follows just what the macroscopically flat surface shows - the ΔG values are only 10 percent higher than kT when considering the smallest difference in the upper and lower bounds on the uncertainties, and only 1.5 times larger when considering the largest difference. This difference is hardly remarkable - and we knew that already from considering that the Kelvin effect is negligible for this size fraction. So there would be a lack of new physical insights, despite some enormous effort, which is indeed regrettable. It seems to this reviewer that an in principle exciting method has been applied to an uninteresting problem in an unconvincing fashion.

Reviewer #2 (Remarks to the Author):

This communication by Qian et al. describes first SFS scattering experiments with airborne aerosol particles. The manuscript describes the surface analysis of water particles decorated with fatty acid molecules. The most important result is that one can do such an experiment. The ability to probe aerosol particles in situ will open up new ways of doing experiments in physical chemistry, biology, material science. The authors also find that the surface propensity of propanoic acid is lower for water droplets than at the air water interface of a bulk solution.

This is a very timely article, well written and useful, functional illustrations. The claims are fully supported by the data. In a way this is a perfect communication in chemistry - a short notice that such experiments can be done with lots of more experiments that can be done by the Rao group or others in the field. I highly recommend publication of the work.

A few points for the authors to address/comment on:

1. In the abstract and the conclusions the authors state they have done "high-performance in situ surface-specific vibrational sum frequency scattering (VSFS)". I am not sure the word "high performance" is needed here. It's also a bit unclear what performance is here.
2. The authors state that the method can be used for in-door measurements with COVID-loaded aerosols. Why only indoor?
3. COVID is mentioned several times as a motivation to measure aerosols. I fully understand that that this is on the authors' mind under the current circumstances. I would, however, recommend to refer to viruses in general here. I predict the paper will be read still in several years when the repeated reference to the pandemic may read a bit odd.
4. The signal intensity in SFS depends on both, the number density of molecules at the interface and their order/orientation. The authors should include the possibility that the order or binding geometry is different at the particle surfaces and not the surface binding affinity.

Reviewer #3 (Remarks to the Author):

This is an excellent paper and I was excited and interested at every step. The authors have shown convincingly that they are able to report on the surface of sub 100 nm aerosols of propionic and butanoic acid in water/NaCl mixtures using sum frequency scattering (VSFS). A comparison to hyper Raman scattering (HRS) for solution phase information was made. Clearly the Langmuir turnover for the VSFS is apparent. I think the authors should comment on the turnover such that the signal becomes further depressed. There may be evidence for dimers and cancelation of signal at the highest concentration. There were only a few grammatical errors, so another thorough read is important. It is interesting that the theoretical attempt to explain the polarization dependence was not successful, although I think it is ok for this to be furthered in the next paper.

There was a sentence, 'Such differences might come from random orientations of propionic acid molecules in the particle phase' around page 6 that referred to the reasoning of the difference in intensities of the HRS and the VSFS. I was a little surprised not to find more information here. Clearly there is a combination of infrared and Raman selection rules for SFS, but the HRS should only have Raman selection rules and of course linear adding of component bands. I believe this deserves several sentences of clarification. Otherwise the readers will incorrectly oversimplify these results. These are relatively minor points, and the paper is ready to publish in my view after these things are addressed. Exciting work.

Responses to Reviewer 1:

Reviewer 1 comment: The authors report what could be an interesting technical result if they had experimental proof that their signal is indeed due to SFS from aerosol particles. Yet, given the small differences in the isotherms for the two systems studied, which is expected from consideration of the Kelvin effect, this reviewer is not convinced that even if the data presented were proven to be indeed due to SFS from aerosol particles, it would provide new physical insights.

Authors' response: Over the 10-year journey to this milestone work the authors have done many control experiments to make sure that the VSFS signal strictly comes from the surface of aerosol particles. The following is one of such control experiments.

To ensure that the signals from the spatially and temporally overlapped 1025 nm and IR pulses were in fact from organic species at the surface of aerosol particles, spectra were collected from aerosols generated from a seed solution only containing 0.5 M NaCl dissolved in ultrapure water (Figure R1 black trace), and from a 4 M solution of propionic acid in the 0.5 M NaCl solution (Figure R1 red trace). The obvious differences displayed in this figure corroborate the author's claims of observing VSFS signals from the surface of aerosol particles.

Figure R1. VSFS spectra of 0.5 M NaCl (black) and 4 M propionic acid in 0.5 M NaCl (red).

While the Kelvin effect is very interesting and could provide much insight to the experimental results in this manuscript, demonstration and investigation of this phenomenon was not the goal of this study. The purpose of this work was to demonstrate that interfacial chemical probing of aerosol particles in the air is possible, and indeed demonstrate differences between this system and its planar analogue. The authors are confident that differences between the planar liquid and curved aerosol interfaces do exist and are significant. Conducting concentration isotherm experiments for other systems such as butanoic acid and ethanol (Figs. R2 and R3, respectively) further support the authors' statements in the manuscript. These

interfacial differences are discussed in more detail below in response to this reviewer's comment regarding ΔG .

Figure R2. Comparison of VSFS, HHH-polarized spectra (blue) and VSFG, PPP-polarized spectra (red) of butanoic acid in 0.5 M NaCl seed solutions. Peak intensity from the $-\text{CH}_3(\text{as})$ mode.

Figure R3. Comparison of VSFS, HHH-polarized spectra (red) and VSFG, PPP-polarized spectra (blue) of ethanol in 0.5 M NaCl seed solutions. Peak intensity from the $-\text{CH}_3(\text{as})$ mode.

The authors have omitted even the most basic controls to show that their signal could be from aerosol particles. If this chamber is the one that was used in the authors' prior SHG work, then the long focal lengths of their setup could easily lead to largely dominating (or even sole) contributions from the optical elements in the chamber. This lack of basic controls indicates the work is highly premature.

Authors' response: The authors believe that after working with laboratory-generated aerosol particles for ten years and with nonlinear optics for even more, that while such controls were performed for each experiment, they have been exhausted in their previous work in this field: *Anal Chem* **2018**, *90*, 10967-10973., *J Phys Chem Lett* **2020**, *11*, 6763-6771., *J Phys Chem A* **2019**, *123*, 6304-6312., and *J Phys Chem Lett* **2016**, *7*, 2294-2297., and that it would be unnecessary to include such redundancies in this manuscript. Unlike traditional environmental chambers, the sole purpose of the aerosol chamber was to prevent possible diffusion of aerosol particles into the greater lab area for safety. Please see Figures R4 & R5 for precise details regarding this chamber. The authors would also like to emphasize, as demonstrated in the schematics below, that when there is now flow of aerosol particles, the 1025 nm and IR pulses are only focused and overlapped in free space.

Figure R4. Schematic diagram of HRS aerosol chamber with IR pulse in blue, 1025 nm pulse in red, and generated HRS signal in green.

Figure R5. Schematic diagram of VSFS aerosol chamber with IR pulse in blue, 1025 nm pulse in red, and generated VSFS signal in green.

Optical components were inspected daily to ensure that there was no accumulation of aerosol residues. Before any VSFS or HRS experiment was conducted, a background spectrum was collected with no flow of aerosol (Figure R13), serving as a check for the CCD, and to ensure no stray signals were being generated by the optical components. To ensure that signals collected by the detection system originated from organics present at the surface of aerosol particles and solely from these species, spectra were collected from aerosols generated from propionic acid in 0.5 M NaCl solution (Figure R1 red), and from 0.5 M NaCl solution (Figure R1 black). Additionally, every concentration isotherm experiment began with a 0 M analyte (0.5 M NaCl) blank seed solution from which aerosols were generated.

Below are points needed for a convincing story:

There is almost no experimental detail given, except for the laser system, detector, and lenses, so that needs to be provided, with detailed drawings. What is the sample chamber composed of? What are the exact dimensions? Where are the windows? How thick and large are they? How is the exhaust connected? How is it guaranteed and validated that no material builds up in the chamber and on the windows?

Authors' response: Thank this reviewer for highlighting these points. Please see the following as an addendum to the experimental section of the manuscript.

Light Paths

After exciting the OPA, the fs IR pulse was directed through a half-waveplate (ThorLabs) followed by a Ge filter (ThorLabs) to remove any residual signal pulse from the OPA. The IR pulse was focused using a 1" CaF₂ lens (f : 15 cm, ThorLabs). The focused IR pulse was spatially overlapped with the 1025 nm pulse at the aerosol flow and was blocked thereafter to prevent any unwanted reflections after the sample.

The 1025 nm pulse was directed to a translation stage after exiting the OPA to control its temporal overlap with the IR pulse. The 1025 nm pulse was then incident on a half-waveplate (ThorLabs) before passing through an air-spaced etalon (SLS Optics) which converted the fs pulse to a ps pulse, providing spectral resolution of 8 cm⁻¹. A 1" lens (f : 25 cm) was used to focus the 1025 nm pulse through a 1000 nm long-pass (ThorLabs) and 1001±117 nm band-pass (BrightLine) filters, mounted into the side panel of the aerosol chamber. The focused 1025 nm was then incident the flow of aerosol particles. A small blocker was placed directly after the aerosol particle stream to ensure neither incident light propagated downfield to the detector.

Generated VSFS and HRS signals were collected independently by 2" lenses (f : 3.2 cm) oriented as shown in Figures R4 & R5 (Figure 1A in manuscript) before exciting the aerosol chamber through holes cut in the side panels. Polarizations for VSFS and HRS signals were selected using polarizing films. Separate 2" lenses (f : 7.5 cm) were used to focus the VSFS and HRS signals into the slit of their respective spectrometers (Acton 300i, Princeton Instruments). The spectrometers were equipped with a charge-coupled device (Princeton Instruments, LN/CCD-1340/400) for spectral collection, controlled with WinSpec software.

Aerosol Handling

Aerosol particles were generated using a constant output pneumatic aerosol generator (TSI 3076) using 40 psi N₂, held at a constant flow of 4 slpm by utilizing a mass flow controller (MKS Instruments) placed before the aerosol generator. The particle density was estimated to be 3.8×10⁶ particles/cm³, with 40 nm central diameter and a broad distribution of 10 to 300 nm in diameter. Particle characterization was conducted using an optical particle sizer and a nanoparticle sizer (TSI 3330, TSI 3910). The experimental details for aerosol particle characterization can be found in the supporting information accompaniments of *J Phys Chem Lett* **2016**, 7, 2294-2297. and *J Phys Chem Lett* **2020**, 11, 6763-6771.

Particles traveled through a 1/4" OD × 11/64" ID PTFE tube to reach the aerosol chamber before being emitted by a cylindrical glass tip with the same diameters. Total distance traveled was measured to be 1 m before reaching incident lasers. The aerosol chamber (Figures R4 & R5) consists of a five-sided box with light-blocking polycarbonate panels and an aluminum frame mounted to the optical table. See Figures R4 & R5 for details on laser inlets and signal outlets. Aerosol flow was introduced through the top panel via a PTFE fitting. Evacuation of aerosols was conducted through a glass catch vessel (Figure R4) placed directly under the flow of aerosols, connected to a vacuum pump.

Chemicals

0.5 M NaCl was prepared using NaCl salt (Fisher Chemical) which had been baked for 10 hours at 600 °C to remove any potential organic contaminants and was dissolved in ultrapure 18.2 M Ω -cm water (Milli-Q, MD Millipore). Propionic and butanoic acids (Acros Organics) were used as received. Fatty acid solutions were prepared in 250 mL volumes and transferred into the 1 L bottle which was supplied with the TSI aerosol generation system. Solutions were continuously stirred using a magnetic stirrer throughout all experiments to ensure sample homogeneity.

There are more fundamental questions regarding the reported results:

Were there any concentration dependent lineshape changes in the VSFS and HRS signal?

Authors' response: Figures R6 and R7 show the VSFS spectra of propionic and butanoic acids, respectively, at different concentrations, and the HRS spectra of propionic acid are shown in Figure R8. These spectra demonstrate that there was no change in spectral lineshape with changing concentration. These spectra are slightly different from those shown in Figure 2 of the manuscript because a different time delay was used during isotherm experiments to preserve brevity. However, using this different delay does not affect the peak position in the spectra. No HRS signal was obtainable for butanoic acid aerosols due to low solubility. The sensitivity of HRS was much lower than that of VSFS. The lack in the change of spectral lineshape with concentration further demonstrates that there was no accumulation of aerosol matter on the optical components, even after exposures of over two hours.

Figure R6. VSFS spectra of propionic acid in 0.5 M NaCl at 1, 4, 7, and 10 M at HHH polarization.

Figure R7. VSFS spectra of butanoic acid in 0.5 M NaCl at 0.15, 0.3, 0.4, and 0.6 M at HHH polarization.

Figure R8. HRS Spectra of propionic acid in 0.5 M NaCl at 1, 4, 7, and 10 M at HHH polarization.

Were HRS bulk signals comparably low for butanoic acid solutions - neither one of these molecules should be very surface active, given their K_{sp} 's?

Authors' response: HRS spectra from the bulk of butanoic acid aerosols were not visible in the experiments. The authors attribute this to the fact that HRS is not a particularly sensitive technique, and to the low solubility of butanoic acid in water. While butanoic acid is obviously somewhat surface active, since we were able to detect it at the planar air/water and gas/aerosol particle interfaces, it is not as surface active as surfactants. The authors commend the reviewer on this observation.

Is there any reason for the nonmonotonic trend in VSFS intensity for the propionic acid samples around 3M concentration?

Authors' response: The authors are very intrigued by this phenomenon and have not yet developed a complete understanding. Based on previous works observing planar interfaces through VSFG, *J. Phys. Chem. B.* **2005**, *109*, 8053-8063. and *J. Phys. Chem. B.* **2005**, *109*, 8064-8075., this decrease could be caused by a number of things, including the formation of a secondary monolayer or a shift in interfacial molecular orientation. Plans for this investigation include conducting VSFS concentration isotherms at a polarization combination which is orientation insensitive.

Were any other spectra taken at different polarizations for the butanoic acid - this section generally could use some more explanation about the discrepancies encountered.

Authors' response: Please see the eight polarization combinations below in Figures R9 and R10. More information on polarization resolved VSFS spectra for butanoic acid was not given as it pertains to interfacial orientation and the scope of this manuscript has been to demonstrate that VSFS signals can be obtained from butanoic acid. The authors are still working on developing and applying a theory to provide interfacial orientation for aerosol particle surfaces.

Figure R9. VSFS spectra of 0.6 M butanoic acid in 0.5 M NaCl at HHH, HHV, HVH, and HVV polarization combinations.

Figure R10. VSFS spectra of 0.6 M butanoic acid in 0.5 M NaCl at VVV, VVH, VHV, and VHH polarization combinations.

What are the particle concentrations and sizes for the different systems, and how were those quantified?

Authors' response:

Figure R11. Aerosol particle number density versus size distribution (diameter in nm) for aerosol particles generated from 0.5 M NaCl.

The concentration of aerosol particles was estimated to be about 3.8×10^6 particles per cubic centimeter. The size distribution of these particles was centered near 40 nm and spanned 10-300 nm as shown in Figure R11.

What are the uncertainty estimates on the adsorption isotherms, esp. in Fig 4A?

Authors' response:

We have adjusted Figures 4A and 4B by adding error bars as shown below.

Figure R12. (A) VSFS electric fields of CH_3 -as for aerosol curved surfaces (red solid circle) and VSFG electric fields of CH_3 -as for the air/water planar interface (blue solid circle) as a function of propionic acid concentration in a 0.5 M NaCl seed solution. (B) VSFS electric fields of $-\text{CH}_3$ -as as a function of butanoic acid concentration in 0.5 M NaCl seed solution (black solid circle), in a comparison with those for propionic acid (red solid circle). Error bars represent 5% relative error.

The key control needed is the one that shows the SFS signals appearing and completely disappearing when the aerosol flow is turned on and off. Even then, additional controls are needed to show zero material uptake on the optical elements in the focused portion of the time- and space-overlapped beam paths.

Authors' response:

Figure R13. VSFS background spectrum collected with no aerosol sample passing through 1025 nm and IR paths.

As stated above, before every experiment began a background spectrum was collected with no flow of aerosol into the chamber, shown in Figure R13. This background shows no signal. As the 1025 nm and IR pulses are only focused and overlapped in the stream of aerosol particles (Figures R4 & R5), there is no signal generated from a different medium. Furthermore, the blocking plate placed directly after the beams prevents propagation or reflection of these pulses.

The second crucial control is the number density dependence of the SFS signal intensity. The authors know how important these are from their pioneering work on particles at Columbia - it's not clear why these important controls and tests were omitted here.

Authors' response: SHS intensity, as shown in *J Phys Chem Lett* **2016**, 7, 2294-2297., is linearly proportional to aerosol particle number density. The authors have assumed that the similar second order nonlinear process of VSFS follows the same trend. However, the authors have assumed for this study that that number density of aerosol particles does not change significantly for the presented systems. A quantitative assessment of this property has not yet been performed as the low signal level of VSFS spectra makes it very difficult. Investigating this phenomenon is planned for future experiments as the authors continue to improve the signal-to-noise ratio.

Figure R14. SHS intensity versus aerosol particle number density. See Supporting Information of *J Phys Chem Lett* **2016**, 7, 2294-2297 for the original graphic.

Even if all of these questions are eventually answered, and even if proof of true aerosol SFS is provided, the ultimate problem with the work is that the new physical insight is unfortunately lackluster in that the adsorption of the two probe species follows just what the macroscopically flat surface shows - the delta G values are only 10 percent higher than kT when considering the smallest difference in the upper and lower bounds on the uncertainties, and only 1.5 times larger when considering the largest difference. This difference is hardly remarkable - and we knew that already from considering that the Kelvin effect is negligible for this size fraction. So there would be a lack of new physical insights, despite some

enormous effort, which is indeed regrettable. It seems to this reviewer that an in principle exciting method has been applied to an uninteresting problem in an unconvincing fashion.

Authors' response:

The authors regret that the reviewer does not see the impacts and applicability of the present work. Due to the relationship between the free energy of adsorption and the measured equilibrium constant, $\Delta G \propto -\ln K$, differences in free energy values between the planar and curved interfaces may appear insignificant. In fact, while the surface adsorption free energies of propionic acid at the air/water interface, -15.38 ± 0.11 kJ/mol, and the aerosol surface, -12.69 ± 0.28 kJ/mol, are hardly more than 20% different on average, the measured surface adsorption constant for the air/water interface, $4.95 \pm 0.53 \times 10^2$, is nearly three times that for the aerosol surface, $1.67 \pm 0.48 \times 10^2$. Additionally, the adsorption constant for butanoic acid at the aerosol surface is $6.26 \pm 0.15 \times 10^2$, which gives an adsorption free energy of -15.96 ± 0.24 kJ/mol, is nearly four times that for propionic acid, indicating that the two fatty acids behave very differently at the surface of aerosol particles. The authors believe that these results are significant and impactful to a variety of fields which are intertwined with highly curved surfaces.

Moreover, the problem which is addressed in this manuscript is that there has yet, to the best of the author's knowledge, to be employed an interface-specific method for the in situ vibrational characterization of the gas/aerosol particle interface. The unique chemical and physical properties of aerosol particle and small droplet surfaces such as viscosity, partitioning rates, surface tension, pH, and charging evoke interesting atmospheric, industrial, biological, and physical implications. (*Environ Sci Technol*, 2015. **49**(19): p. 11485-91., *Nat Commun*, 2014. **5**: p. 3335., *J. Phys. Chem. C.*, 2014. **119**(2): p. 997-1007., *Nat Commun* **2018**, *9*, 956. *Small* **2009**, *5*, 1149-1152., *ACS Nano* **2020**, *14*, 10944-10953., *Annu Rev Phys Chem* **2012**, *63*, 471-491., *Annual Review of Physical Chemistry* **2020**, *71*, 31-51.). The demonstration in this manuscript of VSFS as a method for directly probing vibrational phenomena at the surface of aerosol particles as they float in the air provides an alternative to methods which only observe whole aerosol species, are not conducted under standard conditions, are multiphase in nature, or possess a combination of these flaws (*Environ Sci Technol*, 2019. **53**(24): p. 14441-14448., *Earth and Space Chem*, 2018. **2**(12): p. 1323-1329., *Proc Natl Acad Sci U S A*, 2012. **109**(26): p. 10228-32., *Environ Sci Technol* **2021**, *55*, 14370-14377., *Chem Sci*, 2020. **11**(48): p. 13026-13043., *Annual Review of Physical Chemistry* **2020**, *71*, 31-51., *Angew. Chem. Int. Ed.* **2016**, *55*, 12960-12972., *J. Phys. Chem. A* **2016**, *120*, 2268-2273., *Comm. Chem.* **2019**, *2*. *Phys. Chem. Chem. Phys.* **2019**, *21*, 12434-12445., *Langmuir* **2018**, *34*, 9307-9313.). Finally, even if the differences displayed between aerosol particles and their planar analogues is deemed insignificant, this manuscript fulfills its goal of demonstrating aerosol particles can be probed in situ through VSFS.

Responses to Reviewer 2:

Reviewer 2 comment: This communication by Qian et al. describes first SFS scattering experiments with airborne aerosol particles. The manuscript describes the surface analysis of water particles decorated with fatty acid molecules. The most important result is that one can do such an experiment. The ability to probe aerosol particles *in situ* will open up new ways of doing experiments in physical chemistry, biology, material science. The authors also find that the surface propensity of propanoic acid is lower for water droplets than at the air-water interface of a bulk solution.

This is a very timely article, well written and useful, functional illustrations. The claims are fully supported by the data. In a way this is a perfect communication in chemistry - a short notice that such experiments can be done with lots of more experiments that can be done by the Rao group or others in the field. I highly recommend publication of the work.

A few points for the authors to address/comment on:

1. In the abstract and the conclusions the authors state they have done "high-performance *in situ* surface-specific vibrational sum frequency scattering (VSFS)". I am not sure the word "high performance" is needed here. It's also a bit unclear what performance is here.

Authors' response: The authors are thankful for this reviewer's critical review of the manuscript and are grateful for their appreciation of the findings presented. The authors have tried their best to improve the manuscript. The term "high-performance" was meant to emphasize the low number density of the aerosol particles but has been removed from the manuscript as to not detract from the purpose of the work.

In a revised version, we have made the following changes:

- "Here we demonstrate high performance *in situ* surface-specific vibrational sum frequency scattering (VSFS) to directly identify chemical structures of molecules at aerosol particle surfaces." has been changed to "Here we demonstrate *in situ* surface-specific vibrational sum frequency scattering (VSFS) to directly identify chemical structures of molecules at aerosol particle surfaces."
- "In this work, we present direct probing of chemical species at surfaces of laboratory-generated aerosol particles in real time by developing high-performance vibrational sum frequency scattering (VSFS) technique." has been changed to "In this work, we present direct probing of chemical species at surfaces of laboratory-generated aerosol particles in real time by developing vibrational sum frequency scattering (VSFS) spectroscopy."
- "We have presented *in situ* analysis of chemical structures of organic species at aerosol surfaces by developing a high-performance vibrational sum frequency scattering technique." has been changed to "We have presented *in situ* analysis of chemical

structures of organic species at aerosol surfaces by developing and implementing vibrational sum frequency scattering spectroscopy.”

- “The *in-situ* high-performance VSFS method offers the first approach for understanding the role of aerosol particle surfaces in real time.” has been changed to “The *in situ* VSFS method offers the first approach for understanding the role of aerosol particle surfaces in real time.”

2. The authors states that the method can be used for **in-door** measurements with COVID-loaded aerosols. Why only indoor?

Authors’ response: Thank this reviewer for pointing it out. We agreed with him or her on that it is essential to characterize the samples we used. The method described in the manuscript could potentially be used for outdoor aerosols as well as indoor species. The authors have accepted this suggestion as follows:

- “Our approach opens a new avenue in revealing surface compositions and chemical aging in the formation of secondary organic aerosols in the atmosphere as well as chemical analysis of in-door COVID-19 aerosol particles.” has been changed to “Our approach opens a new avenue in revealing surface compositions and chemical aging in the formation of secondary organic aerosols in the atmosphere as well as chemical analysis of indoor and outdoor viral aerosol particles.”
- Furthermore, the chemical identifications at the aerosol surfaces also have significant implications in the detection of indoor transmission of aerosol particles.” has been changed to “Furthermore, the chemical identifications at the aerosol surfaces also have significant implications in the detection of indoor and outdoor transmission of aerosol particles.”

3. COVID is mentioned several times as a motivation to measure aerosols. I fully understand that that this is on the authors mind under the current circumstances. I would, however, recommend to refer to viruses in general here. I predict the paper will be read still in several years when the repeated reference to the pandemic may read a but odd.

Authors’ response: Please thank the reviewer for this great observation.

- “Of special interest are aerosol particles floating in the air for both indoor COVID-19 transmission and atmospheric chemistry.” has been changed to “Of special interest are aerosol particles floating in the air for both indoor virus transmission and atmospheric chemistry.”
- “Our approach opens a new avenue in revealing surface compositions and chemical aging in the formation of secondary organic aerosols in the atmosphere as well as chemical analysis of in-door COVID-19 aerosol particles.” has been changed to “Our approach opens a new avenue in revealing surface compositions and chemical aging in

the formation of secondary organic aerosols in the atmosphere as well as chemical analysis of indoor and outdoor viral aerosol particles.”

4. The signal intensity in SFS depends on both, the number density of molecules at the interface and their order/orientation. The authors should include the possibility that the order or binding geometry is different at the particle surfaces and not the surface binding affinity.

Authors' response: Thank this reviewer for pointing it out. The authors acknowledge that sum frequency intensity is dependent on both surface orientation and number density. However, as the current theory of VSFS for orientation at aerosols is incomplete, the primary goal is to resolve this issue. Once the authors can quantify molecular orientation at aerosol surfaces, the authors plan to conduct concentration dependent VSFS experiments under a polarization combination which is insensitive to orientation, allowing the investigation of surface number density by itself.

Responses to Reviewer 3:

Reviewer 3 comment: This is an excellent paper and I was excited and interested at every step. The authors have shown convincingly that they are able to report on the surface of sub 100 nm aerosols of propionic and butanoic acid in water/NaCl mixtures using sum frequency scattering (VSFS). A comparison to hyper Raman scattering (HRS) for solution phase information was made. Clearly the Langmuir turnover for the VSFS is apparent. I think the authors should comment on the turnover such that the signal becomes further depressed. There may be evidence for dimers and cancelation of signal at the highest concentration. There were only a few grammatical errors, so another thorough read is important. It is interesting that the theoretical attempt to explain the polarization dependence was not successful, although I think it is ok for this to be furthered in the next paper.

There was a sentence, 'Such differences might come from random orientations of propionic acid molecules in the particle phase' around page 6 that referred to the reasoning of the difference in intensities of the HRS and the VSFS. I was a little surprised not to find more information here. Clearly there is a combination of infrared and Raman selection rules for SFS, but the HRS should only have Raman selection rules and of course linear adding of component bands. I believe this deserves several sentences of clarification. Otherwise the readers will incorrectly oversimplify these results. These are relatively minor points, and the paper is ready to publish in my view after these things are addressed. Exciting work.

Authors' response: Thank the reviewer for the encouraging comments. The three peaks shown in the HRS spectrum of propanoic acid in 0.5 M NaCl are all Raman-active. We agree with this reviewer on that some difference between the two spectra were expected. VSFS is a combination of infrared and Raman. On the other hand, the HRS should only have Raman selection rules. We were also surprised that there were no changes in HRS spectra regarding peak position but just relative intensity. The authors do not, however, rule out the possibility of these modes with both IR and Raman active. The IR contributions could be minor in these spectra. As the authors continue to utilize this method of simultaneous VSFS and HRS for aerosols containing different organic species they hope to gain more insight into these vibrational phenomena from the aerosol particle bulk.

Reviewers' comments:

Reviewer #1 (Remarks to the Author):

This reviewer appreciates the efforts made by the authors to improve their manuscript. The additional information is certainly not hurting the present state of the work. Yet, the key experiment is not provided: the functional dependence of the SFG intensity in the particle number density.

The authors state they lack the sensitivity to carry out this study, but the data say otherwise: The lineshapes are clear and the S/N is good. There really is no need to rush this work without this key control. It would be a shame if the ISFG vs Nparticles control did not work out the way one would hope.

This reviewer also appreciates the enthusiasm by the other reviewers, at least in round one of the review process, but needs to point out that the necessary control is going to be beneficial for the entire field as it will prove that what the manuscript claims is indeed correct. Again, it would be a shame if it were not.

Regarding the comments about the differences in the K_{eq} 's for the two systems studied: those are really miniscule differences that are quite insignificant at room temperature thermal energy. What counts is the ΔG , which shows no differences beyond a couple of kJ/mol at most. Therefore, the original conclusion about a lack of new physical insights in this manuscript remains the same.

Responses to Reviewer #1

This reviewer appreciates the efforts made by the authors to improve their manuscript. The additional information is certainly not hurting the present state of the work. Yet, the key experiment is not provided: the functional dependence of the SFG intensity in the particle number density.

The authors state they lack the sensitivity to carry out this study, but the data say otherwise: The lineshapes are clear and the S/N is good. There really is no need to rush this work without this key control. Its would be a shame if the ISFG vs Nparticles control did not work out the way one would hope.

This reviewer also appreciates the enthusiasm by the other reviewers, at least in round one of the review process, but needs to point out that the necessary control is going to be beneficial for the entire field as it will prove that what the manuscript claims is indeed correct. Again, it would be a shame if it were not.

Authors' Response:

The authors appreciate this reviewer reception of the authors' responses. The authors have conducted VSFS experiments to investigate the relationship between signal intensity and aerosol particle density, as shown in Figure R1. By diluting the 4 slpm flow of aerosol particles with an interjecting flow of humidified N₂ gas controlled through an additional mass flow controller, we see a linear relationship between VSFS signal intensity and aerosol particle density. Figure R2 shows the spectra which were collected for this experiment. As is expected, sum frequency scattering signals from aerosol particle surfaces are proportional to particle density, as they are for second harmonic scattering spectroscopy (*J Phys Chem Lett* **2016**, 7, 2294-2297). The manuscript has also been revised to include this experiment, with its graphical results shown in a new Supporting Information file (see attached).

Figure R1. Relationship of aerosol particle density to VSFS intensity. Error bars represent 1 standard deviation.

Figure R2. VSFS spectra at different particle densities, 0-100 % at HHH Polarization.

Regarding the comments about the differences in the K_{eq} 's for the two systems studied: those are really miniscule differences that are quite insignificant at room temperature thermal energy. What counts is the ΔG , which shows no differences beyond a couple of kJ/mol at most. Therefore, the original conclusion about a lack of new physical insights in this manuscript remains the same.

Authors' Response:

The authors regret that the reviewer does not see the impacts and applicability of the present work. Again, the measured surface adsorption constant for the air/water interface, $4.95 \pm 0.53 \times 10^2$, is nearly three times that for the aerosol surface, $1.67 \pm 0.48 \times 10^2$. Additionally, the adsorption constant for butanoic acid at the aerosol surface is $6.26 \pm 0.15 \times 10^2$, which gives an adsorption free energy of -15.96 ± 0.24 kJ/mol, is nearly four times that for propionic acid, indicating that the two fatty acids behave very differently at the surface of aerosol particles.

Again, due to the relationship between the free energy of adsorption and the measured equilibrium constant, $\Delta G \propto -\ln K$, differences in free energy values between the planar and curved interfaces may appear insignificant.

The authors continue to believe that these results are significant and impactful to a variety of fields which are intertwined with highly curved surfaces.

REVIEWERS' COMMENTS:

Reviewer #1 (Remarks to the Author):

The authors have taken the most important suggestion seriously and provided the experimental proof that their signal is truly SFS. Given that this result is so important, this reviewer strongly encourages the authors to include the SFS vs particle density result into the main text. The authors are also encouraged to emphasize that the differences in the Kads values are such that the corresponding differences in deltaG values between the flat and the particle process, for the molecules studied, are in most cases not statistically significant. Once those changes are made, this reviewer is happy to recommend publication.

Response to Reviewer 1:

Reviewer #1 (Remarks to the Author):

The authors have taken the most important suggestion seriously and provided the experimental proof that their signal is truly SFS. Given that this result is so important, this reviewer strongly encourages the authors to include the SFS vs particle density result into the main text. The authors are also encouraged to emphasize that the differences in the Kads values are such that the corresponding differences in deltaG values between the flat and the particle process, for the molecules studied, are in most cases not statistically significant. Once those changes are made, this reviewer is happy to recommend publication.

Author's Response: Please thank the reviewer for accepting the results provided previously. The authors have added the results of the particle density vs VSFS intensity to the main text in Figure 2c. The revised Figure 2 shown in the edited manuscript is shown below.

Figure 2: Spectra and isotherms of VSFS and HRS.

a HHH-polarized configuration spectra for VSFS (upper) and HRS (lower) for aerosol particles from a 0.5 M NaCl seed solution with 4.0 M propionic acid. **b** VSFS (red solid circle, left axis) and HRS (blue solid triangle, right axis) intensities at 2991.8 cm^{-1} ($\text{CH}_3\text{-as}$) as a function of propionic acid concentrations added. Error bars represent 5% deviation. **c** Relationship of aerosol particle density to VSFS intensity. Error bars represent 1 standard deviation.

The authors have also accepted the suggestion of discussing the statistical insignificance in ΔG differences for the adsorption of propionic acid to aerosol particle and planar liquid surfaces. The penultimate paragraph of the Results and Discussion section of the manuscript has been edited to read as follows:

We first compared the surface adsorption free energy of propionic acid at the aerosol surface with that at the air/water interface. Fittings of the model in Eq. 1 to the two curves in Figure 4a generate the surface adsorption constants for propionic acid of $1.67 \pm 0.48 \times 10^2$ at the aerosol surface, and $4.95 \pm 0.53 \times 10^2$ at the air/water interface, giving adsorption free energy values of $-12.69 \pm 0.28 \text{ kJ mol}^{-1}$ and $-15.38 \pm 0.11 \text{ kJ mol}^{-1}$, respectively. It is surprising that the adsorption ability of the molecules at the curved particle surface is lower than that at the planar surface. Although the large difference in adsorption constants is effectively lost in the logarithmic free energy function for propionic acid, this difference may be more dramatic when considering other adsorbates.